# Do Large Language Models Understand Scientific Argumentation?

## Abstract

Scientific discourse depends on argumentative reasoning: identifying claims, evaluating evidence, and constructing coherent responses. While recent advances in reasoning-capable language models have demonstrated strong performance on mathematical and logical benchmarks, their ability to engage in scientific argumentation remains unclear. We present the first systematic evaluation of language models across eight tasks spanning argument mining, rebuttal generation, and discourse-level reasoning using research papers, peer reviews, and grant proposals. Our study reveals that even frontier models with strong general reasoning skills struggle with domain-specific argumentative tasks, highlighting a fundamental capability gap. To address this, we introduce a training framework that explicitly scaffolds the argumentative reasoning process in language models, substantially improving their competence in scientific discourse. The resulting compact models approach or exceed the performance of much larger proprietary systems and generalize to unseen conversational settings, demonstrating reasoning transfer beyond task-specific supervision. These findings underscore that effective scientific argumentation is not an emergent property of scale, but requires explicit reasoning-aware training, and they point toward practical pathways for building AI systems that can contribute meaningfully to scientific discourse.

## 1 Introduction

Scientific progress fundamentally depends on evidence-based argumentation, as researchers advance claims, marshal evidence, critique methodologies, and synthesize competing perspectives into coherent theories (Toulmin, 1958; Kuhn, 1997; Teufel & Moens, 1999; Driver et al., 2000). This process demands precise reasoning skills, including the ability to trace logical connections between hypotheses and data, evaluate the strength of empirical support, and construct rebuttals that address both methodological and theoretical concerns (Teufel et al., 1999; de Waard & Pander Maat, 2012). As large language models (LLMs) increasingly assist in research tasks, from literature review to manuscript drafting (OpenAI, 2025a; Lu et al., 2024; Yamada et al., 2025; Gottweis et al., 2025; Schmidgall et al., 2025), a critical question emerges: *can these systems engage in the rigorous argumentative reasoning that defines scientific discourse?*

Recent reasoning-focused models achieve impressive results on mathematical and logical benchmarks (OpenAI, 2025b; Guo et al., 2025; Abdin et al., 2025), yet scientific argumentation poses distinct challenges. Unlike proofs or puzzles, it requires domain-specific methodological knowledge, careful evaluation of evidence, and the ability to navigate uncertainty and conflicting interpretations. Existing evaluations largely target mathematics, code, or general logic (Chollet, 2019; Chollet et al., 2025; Balunović et al., 2025), leaving the argumentative demands of science unexplored.

The challenge is particularly acute because scientific discourse differs fundamentally from general argumentation and existing computational argumentation research has largely focused on news articles, social media, or debate transcripts (Slonim et al., 2021; Chen et al., 2019; Roush & Balaji, 2020). While valuable, these domains do not capture the specialized reasoning patterns that characterize scholarly communication. Where social media debates or news commentary may rely on rhetorical persuasion, scientific arguments must satisfy stringent evidential standards. Moreover, scientific reasoning often spans multiple discourse levels, from sentence-level claim detection to

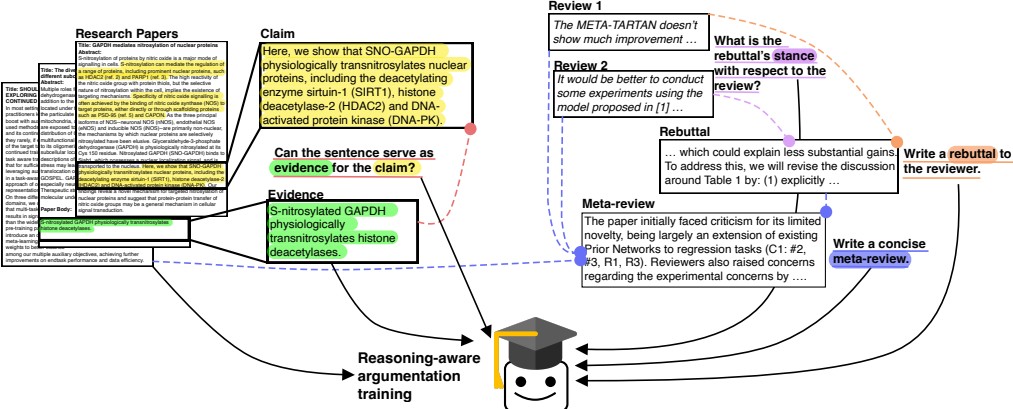

Figure 1: The figure shows eight tasks across argument mining, generation, and discourse-level reasoning, with examples ranging from claim detection in research papers to rebuttal generation addressing reviewer concerns. Our reasoning-aware training approach scaffolds argumentative capabilities through structured supervision across these diverse scientific discourse tasks.

document-level evidence synthesis, requiring models to maintain coherent logical threads across extended contexts.

We address this gap with a comprehensive investigation of scientific argumentation in LLMs. Our contributions are fourfold: (1) we introduce a systematic evaluation framework spanning eight tasks across research papers, peer reviews, and grant proposals; (2) we provide the first capability audit showing that strong general reasoning does not transfer to scientific argumentation; (3) we propose a training methodology that explicitly scaffolds the argumentative reasoning process through distilled chain-of-thought (CoT) supervision and reinforcement learning, yielding substantial improvements; and (4) we release evaluation protocols, reasoning traces, and trained models to support future research.

## 2 DO LLMS UNDERSTAND SCHOLARLY ARGUMENTATION?

### 2.1 TASK FORMULATION

Our evaluation spans three task families covering the full spectrum of scientific argumentation including argument mining and generation, and discourse-level argumentation as shown in Figure 1.

**Argument Mining.** We evaluate four core tasks that assess models' ability to identify and classify argumentative structures in scientific text.

(1) *Claim detection* identifies verifiable scientific claims within NSF grant abstracts using NSF-SciFy (Rao et al., 2025). Given numbered sentences from grant abstracts, models must identify which contain empirically testable claims (e.g., "This approach will reduce computational complexity by 40%") versus background statements or methodology descriptions. The task uses binary classification with claim indices as targets (e.g., "0,2,5" or "none").

(2) *Evidence detection* determines whether sentences can serve as evidence for scientific claims using SciFact (Wadden et al., 2020). Models classify sentence-claim pairs into three categories: SUPPORT (sentence provides evidence supporting the claim), CONTRADICT (sentence provides evidence refuting the claim), or NEI (sentence is irrelevant or insufficient). This dataset is significantly imbalanced towards NEI samples (95% of the labels). Hence, we report F1 score along with accuracy.

(3) *Evidence classification* assigns support relationships using SciFact (Wadden et al., 2020). Given claim-evidence pairs where relevance is established (i.e. no NEI sentences), models classify the relationship as SUPPORTS or REFUTES.

| Model | Claim Detection | | Evidence Detection | | Evidence Classification | | Stance Detection | |
|---|---|---|---|---|---|---|---|---|
| | Acc | $F_1$ | Acc | $F_1$ | Acc | $F_1$ | Acc | $F_1$ |
| Llama3.2 1B | 6.5 | 7.5 | 12.5 | 7.4 | 19.5 | 25.3 | 16.0 | 19.1 |
| Llama3.2 3B | 24.5 | 41.1 | 60.0 | 25.0 | 56.0 | 58.6 | 42.5 | 31.2 |
| Llama3.1 8B | 72.5 | 82.8 | 82.5 | 33.6 | 76.0 | 76.1 | 40.0 | 33.5 |
| OpenAI o3 | **96.0** | **92.0** | **99.0** | **49.8** | 86.0 | **86.5** | **49.5** | **53.3** |
| Gemini 2.5 Flash | 93.5 | 91.0 | 94.5 | 43.5 | **86.5** | 87.0 | 32.5 | 38.9 |

Table 1: Zero-shot performance on argument mining tasks. Despite strong general reasoning capabilities, even frontier models (OpenAI o3) struggle with nuanced tasks like stance detection. The gap between accuracy and F1 reveals failures in handling class imbalance, particularly in evidence detection where models default to majority class predictions.

(4) *Stance detection* analyzes peer review discourse using DISAPERE (Kennard et al., 2022) where the dataset contains reviews from computer science venues. Models classify author rebuttal sentences relative to reviewer comments into three stances: CONCUR (author agrees with or acknowledges the reviewer's point), DISPUTE (author challenges or refutes the reviewer's concern), or NON-ARG (sentence contains social pleasantries or non-argumentative content).

**Argument Generation.** We assess two generation capabilities requiring coherent argumentative discourse.

(1) *Rebuttal generation* using ARIES (D'Arcy et al., 2024) evaluates models' ability to compose scholarly rebuttals addressing reviewer concerns. Given reviewer comments and paper excerpts, models must generate point-by-point responses that acknowledge valid concerns, clarify misunderstandings, and defend methodological choices. The dataset contains reviewer-rebuttal pairs from major AI conferences. Responses are evaluated using ROUGE-L (Lin, 2004), METEOR (Denkowski & Lavie, 2011), and BERTScore (Zhang et al., 2020) against human-authored rebuttals.

(2) *Meta-review summarization* using PRRCA (Wu et al., 2022) requires synthesizing multiple peer reviews into coherent meta-reviews that identify consensus points, highlight disagreements, and provide acceptance recommendations. Models process individual reviews plus author rebuttals to generate meta-reviews that capture reviewer sentiment. The dataset contains review sets from computer science conferences with corresponding meta-reviews averaging 150-200 words.

**Discourse-Level Argumentation.** We evaluate answer composition task that require coherence and evidence integration for discourse-level argumentation.

*Systematic answer composition* using QASA (Lee et al., 2023) evaluates evidence-grounded question answering in scientific contexts. Given research questions and multiple evidence passages, models must synthesize information to produce comprehensive answers that cite supporting evidence and acknowledge limitations. The dataset contains questions from computer science and biomedical literature with gold answers averaging 80-120 words. Questions require multi-hop reasoning across evidence passages, such as "How do attention mechanisms in transformers compare to memory-augmented networks for long-sequence modeling?". We evaluate generated answers by using ROUGE-L, METEOR, and BERTScore against human-written answers.

Each task family represents increasing complexity: argument mining tests recognition of existing argumentative structures, argument generation requires producing coherent argumentative text, and discourse-level tasks demand sustained reasoning across multiple turns or evidence sources while maintaining argumentative coherence and factual grounding.

## 2.2 BASELINES

**Systematic failures across model scales.** Tables 1 and 2 reveal that current LLMs, regardless of size or reasoning capabilities, struggle fundamentally with scientific argumentation. The performance gap between general capabilities and domain-specific tasks is substantial, as models achieving near-perfect mathematical reasoning barely exceed random chance on stance detection. This pattern holds

| Task | Dataset | Method | BERTScore | ROUGE-L | METEOR |
|---|---|---|---|---|---|
| **Generation** | ARIES | Llama3.2 1B | 0.41 | 0.04 | 0.02 |
| | | Llama3.2 3B | 12.69 | 1.81 | 1.29 |
| | | Llama3.1 8B | 33.60 | 3.96 | 2.75 |
| | | OpenAI o3 | **84.29** | **9.90** | **5.55** |
| | | Gemini 2.5 Flash | 55.65 | 12.12 | 13.89 |
| **Summarization** | PRRCA | Llama3.2 1B | 0.84 | 0.12 | 0.08 |
| | | Llama3.2 3B | 79.81 | 12.89 | 13.32 |
| | | Llama3.1 8B | **84.34** | **14.56** | 14.53 |
| | | OpenAI o3 | 83.89 | 14.01 | **21.08** |
| | | Gemini 2.5 Flash | 78.23 | 13.49 | 17.00 |
| **Discourse** | QASA | Llama3.2 1B | 7.63 | 0.35 | 0.17 |
| | | Llama3.2 3B | 24.56 | 2.47 | 1.26 |
| | | Llama3.1 8B | 81.14 | 11.63 | 6.10 |
| | | OpenAI o3 | 78.23 | **13.02** | **9.18** |
| | | Gemini 2.5 Flash | **82.25** | 9.89 | 8.14 |

Table 2: Zero-shot performance on generation and discourse tasks evaluated semantically and lexically. The stark performance drop from summarization to generation tasks indicates models can reorganize but not construct arguments. High BERTScore with low ROUGE-L in discourse tasks suggests grammatical coherence without argumentative substance.

across all task families, suggesting that scientific argumentation requires capabilities beyond what current training objectives provide.

**Class imbalance exposes shallow pattern matching.** The divergence between accuracy and F1 scores, particularly evident in evidence detection tasks where the distribution is significantly skewed towards non-relevant information, indicates models default to majority class predictions rather than understanding evidential relationships. High accuracy with low F1 reveals an inability to distinguish genuine evidence from irrelevant text, which is a critical failure for scientific discourse.

**Generation quality inversely correlates with task complexity.** While models achieve reasonable performance on meta-review summarization (a consolidation task), they fail catastrophically at rebuttal generation (requiring novel argumentation). This gradient suggests models can reorganize existing arguments but cannot construct new ones, as they mimic form without understanding function.

These baseline results demonstrate that scientific argumentation represents a fundamental capability gap in current models, motivating specialized training approaches that explicitly teach argumentative reasoning rather than relying on emergent capabilities from scale.

## 3 TEACHING LLMS SCIENTIFIC ARGUMENTATION

The cognitive progression of human argumentative skill development involves recognizing patterns, producing structured arguments, and generalizing through feedback. Our approach mirrors this natural progression with a three-stage training pipeline that progressively builds argumentative capabilities through reasoning-aware knowledge distillation, multi-task supervised learning, and reinforcement learning with composite rewards.

### 3.1 REASONING-AWARE KNOWLEDGE DISTILLATION

Scientific argumentation demands capabilities that extend beyond surface-level pattern matching. We hypothesize that the reasoning capabilities emerging in recent LLMs can be channeled toward argumentation through explicit supervision of intermediate reasoning steps.

We employ Gemini 2.5 Flash (Team, 2025) as a teacher model to generate structured reasoning traces for our training corpus. The distillation process covers 3,000 examples balanced across tasks, with 500 examples per task category except for SciFact variants, where we limit collection to 250 examples each to prevent dataset skew toward fact-verification patterns. We adopt a reverse-engineering

paradigm where the teacher receives gold labels and constructs post-hoc reasoning chains that justify these answers. This supervised approach ensures high-quality reasoning grounded in correct outcomes, avoiding the generation of plausible but ultimately incorrect logical paths that plague unsupervised reasoning generation. The distillation process produces structured outputs combining explicit reasoning with final answers:

$$\text{output} = \langle\texttt{reasoning}\rangle\mathcal{R}\langle\texttt{/reasoning}\rangle\langle\texttt{answer}\rangle\mathcal{A}\langle\texttt{/answer}\rangle \tag{1}$$

where $\mathcal{R}$ represents 2–4 sentences of grounded analysis and $\mathcal{A}$ contains the target response.

We enforce three critical constraints to ensure reasoning quality. First, reasoning traces must avoid explicit mention of classification labels to encourage semantic understanding over lexical pattern matching. Second, all reasoning must reference the provided context, preventing hallucination and ensuring grounding. Third, we limit reasoning length to maintain focus and prevent the introduction of noise through verbose explanations. These constraints are enforced through explicit instructions in the teacher prompt and validated through post-processing checks.

For long-document tasks such as rebuttal generation and meta-review summarization, we augment the distillation process with BM25-based passage retrieval. Given a document $\mathcal{D}$ and query $q$, we identify relevant spans using standard BM25 scoring with parameters $k_1 = 1.5$ and $b = 0.75$, selecting top-$k$ passages with overlap suppression. This teaches models to work with condensed context which is a critical skill for real-world scientific discourse where documents often exceed model context windows.

## 3.2 MULTI-TASK SUPERVISED FINE-TUNING

The supervised fine-tuning (SFT) phase transforms distilled reasoning traces into learned argumentative capabilities. We fine-tune Llama 3.2 models (Team, 2024) in both 1B and 3B configurations using full parameter updates. The training employs cross-entropy loss over the complete output sequences including reasoning traces, with loss masking applied to system and user tokens to focus learning on model-generated content.

Training proceeds as a multi-task learning problem where models simultaneously acquire argumentative patterns across diverse scientific contexts. We keep the number of samples balanced in this stage to ensure balanced gradient contributions across tasks. This prevents larger datasets from dominating the learning dynamics as we experienced in our early experiments, and ensures comprehensive skill acquisition across all argumentative capabilities.

The supervised phase establishes two critical foundations: structured output generation with consistent XML-style tags enabling clean reasoning extraction, and learned correlations between reasoning patterns and correct answers. However, models trained solely through supervised learning exhibit two fundamental limitations. (1) Performance remains bounded by teacher model quality, creating an artificial ceiling on capability improvement. (2) Supervised learning tends to produce reasoning mimicry which results with syntactically correct but semantically shallow reasoning chains that fail to generalize. These limitations motivate our subsequent reinforcement learning phase, which we analyze through systematic ablations in Section 4.1.

## 3.3 REINFORCEMENT LEARNING WITH COMPOSITE REWARDS

The final training stage employs reinforcement learning to refine argumentative capabilities beyond teacher-bounded performance. We adopt Group Relative Policy Optimization (GRPO) (Shao et al., 2024), generating multiple responses per prompt and optimizing based on relative quality within each group. This approach naturally handles the multi-objective nature of scientific argumentation, where responses must balance correctness, logical coherence, and evidential grounding.

Our reward design explicitly targets the requirements of scientific discourse through four complementary components. Answer correctness $r_{\text{ans}}$ employs task-appropriate metrics: exact match for classification tasks, F1 for multi-label problems, and ROUGE-L for generation tasks. Logical faithfulness $r_{\text{faith}}$ measures consistency across the reasoning chain using natural language inference scores, with weights prioritizing answer grounding (0.5) over intermediate consistency (0.3) and reasoning-answer alignment (0.2). Evidence attribution $r_{\text{attr}}$ quantifies the proportion of reasoning

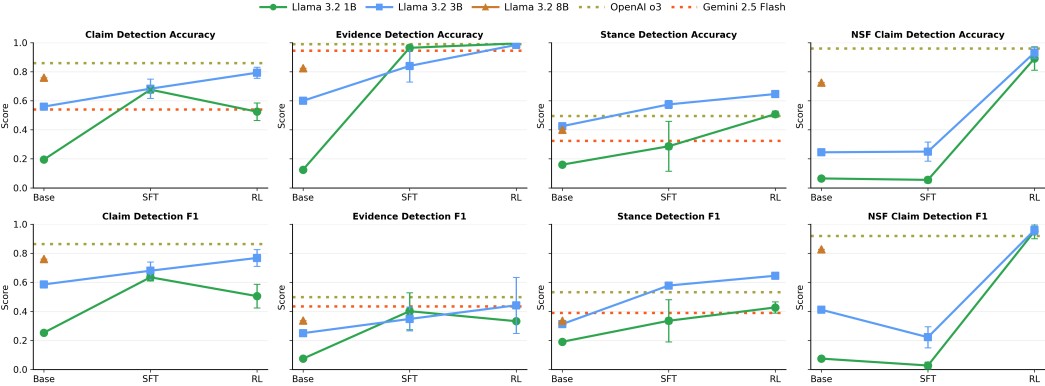

Figure 2: Performance progression across training stages for argument mining tasks. Error bars show standard deviation across three random seeds. The progression from base model to RL demonstrates systematic improvements, with the most substantial gains occurring during the SFT stage. Notably, our trained 3B model surpasses commercial baselines on several tasks despite being orders of magnitude smaller.

tokens traceable to source context, encouraging explicit citation practices essential to scientific writing. Format compliance $r_{\text{fmt}}$ ensures structural validity while penalizing tag spillover that would compromise downstream parsing.

The composite reward incorporates a multiplicative gating mechanism that enforces minimum reasoning standards:

$$r(x, y, g) = \left( \sum_{c \in \mathcal{C}} w_c \cdot r_c(x, y, g) \right) \cdot \text{gate}(y, x) \tag{2}$$

where the gate function combines sigmoid-transformed measures of reasoning length and context coverage:

$$\text{gate}(y, x) = \sigma \left( \alpha \cdot (|\mathcal{R}_y| - \tau) \right) \cdot \sigma \left( \beta \cdot \text{coverage}(\mathcal{R}_y, x) \right) \tag{3}$$

This multiplicative structure ensures models cannot achieve high rewards through correct answers alone, and they must demonstrate adequate reasoning depth and maintain substantial overlap with source material. The parameters $\alpha = 0.2$ and $\beta = 8.0$ were determined through preliminary experiments to balance reasoning quality with generation diversity.

Training incorporates several design decisions critical for generalization. We include 50% held-out prompts unseen during supervised fine-tuning to prevent overfitting to specific reasoning patterns. Discourse-level tasks remain entirely excluded from reinforcement learning, reserved for zero-shot evaluation of emergent capabilities. Adaptive KL penalties prevent reward hacking while maintaining generation diversity, with $\beta$ scheduling responding dynamically to divergence measurements. More details of the training setup explained in C. The impact of each reward component is systematically analyzed through ablation studies in Section 4.1.1, revealing the critical role of answer correctness while highlighting surprising interactions between faithfulness constraints and metric gaming.

## 4 RESULTS & FINDINGS

We evaluate our approach across three independent training runs, with results presented in Figures 2 and 3.

**The primacy of supervised reasoning.** The most notable pattern across all tasks is the transformative impact of supervised fine-tuning with reasoning traces. While baseline models struggle with fundamental tasks, such as achieving near-random performance on claim detection, the introduction of explicit reasoning supervision yields improvements of 150-400% across argument mining tasks

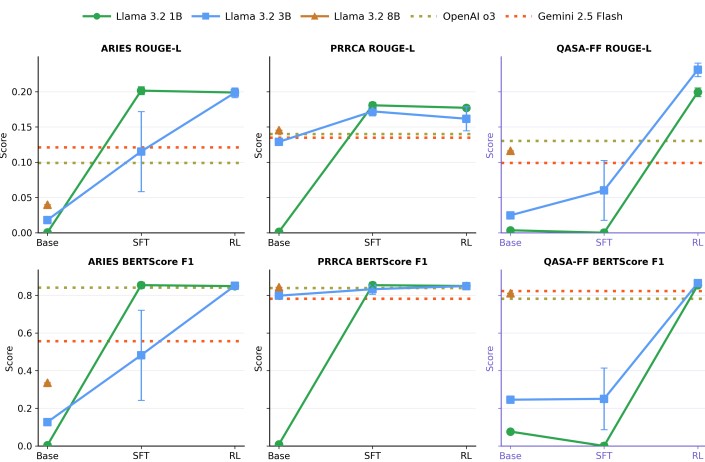

Figure 3: Performance progression across training stages for argument generation and discourse-level tasks. Line colors and error bars follow the same convention as Figure 1. Purple-shaded axes distinguish discourse-level tasks from standard generation tasks. The substantial improvements in BERTScore across all tasks indicate enhanced semantic coherence.

(Figure 2). This dramatic shift suggests that scientific argumentation is not an emergent property of scale but rather a capability that must be explicitly taught through structured reasoning patterns.

The subsequent reinforcement learning stage provides meaningful but more modest gains, typically 10-20% additional improvement. This diminishing return curve indicates that once models acquire the fundamental reasoning framework through supervision, further optimization primarily refines execution rather than introducing new capabilities. Notably, our 3B model after full training surpasses OpenAI o3 on stance detection (63% vs 59.3% accuracy) and approaches its performance on evidence detection, despite being orders of magnitude smaller.

**Task complexity reveals capability boundaries.** Performance patterns across tasks reveal a clear hierarchy of difficulty that illuminates the nature of scientific argumentation. Binary classification tasks (claim detection, evidence classification) show the strongest improvements, with F1 scores reaching 0.92-0.97 after training. Multi-class tasks with nuanced distinctions, particularly stance detection where models must differentiate agreement, disagreement, and non-argumentative content, prove substantially more challenging, with even our best models achieving only 62.4% F1.

Generation tasks (Figure 3) expose an interesting dichotomy. Rebuttal generation shows dramatic improvement (BERTScore: $0.126 \rightarrow 0.850$), approaching commercial model performance. However, meta-review summarization plateaus at lower absolute performance, suggesting that synthesizing multiple perspectives into coherent summaries requires capabilities beyond what current training approaches provide. This gap between generating novel arguments and consolidating existing ones highlights a fundamental limitation in how models understand multi-party scientific discourse.

**Generalization beyond training distribution.** The purple-shaded region in Figure 3 highlights the performance on discourse task and provides critical evidence for capability transfer. While QASA is included in supervised training, it is entirely excluded from reinforcement learning yet shows substantial improvements during the RL stage. The large increase in QASA scores during RL training, despite no direct supervision, indicates our approach develops transferable argumentative reasoning rather than task-specific optimizations.

### 4.1 ABLATIONS

To understand the mechanisms underlying our approach's effectiveness, we conduct ablations of both reward components and chain-of-thought supervision. These studies provide crucial insights into which components drive performance gains and under what conditions. We share our findings on ablations in this section, defer detailed tabular results to A due to spacing constraints.

#### 4.1.1 REWARD COMPONENT ABLATIONS

To ablate the impact of each reward component on the performance of the model, we train the Llama 3.2 1B model for 1.5 epochs with all reward components and then remove one at a time. Table 3 and Table 4 present our reward component ablation results across classification and generation tasks respectively. The analysis reveals distinct component importance patterns that illuminate the mechanisms of reasoning-aware training.

**Answer accuracy reward ($r_{\text{ans}}$) serves as the fundamental optimization signal.** The answer accuracy component ($r_{\text{ans}}$) proves indispensable, with removal causing catastrophic failure across all tasks. This dominance suggests that while reasoning traces improve performance, they function primarily as scaffolding for correct answers rather than independent logical structures. Attribution and format rewards show task-dependent importance, with attribution particularly critical for generation tasks where hallucination poses greater risks.

**Faithfulness reward ($r_{\text{faith}}$) exhibits counterintuitive effects that reveal evaluation limitations.** Table 3 reveals a counterintuitive result: removing faithfulness constraints frequently improves automatic evaluation metrics. For instance, claim detection accuracy increases from 84.75% to 99% in the absence of faithfulness rewards, with evidence detection showing comparable improvements. This paradox highlights a fundamental evaluation challenge: when unconstrained by consistency requirements, models may exploit weaknesses in metrics rather than develop genuine reasoning capabilities. Nevertheless, following prior work (Chen et al., 2025), we maintain faithfulness as a component of the reward signal, since doing so helps reduce logical errors and better align the model's reasoning with its outputs, thereby mitigating potential failure modes.

#### 4.1.2 CHAIN-OF-THOUGHT SUPERVISION ABLATIONS

Tables 5 and 6 reveal that chain-of-thought supervision exhibits strong task-dependency. Complex reasoning tasks show severe degradation without CoT as claim detection drops from 93.5% to 48.25% accuracy, while QASA BERTScore plummets from 86.85 to 43.43. These tasks require explicit logical scaffolding to maintain coherence across multiple inferential steps.

Conversely, tasks with strong surface patterns show minimal CoT dependence. Evidence classification and rebuttal generation perform similarly with or without explicit reasoning, suggesting these tasks rely more on learned templates than multi-step inference. Interestingly, reinforcement learning partially compensates for missing CoT supervision in some tasks, indicating that reward signals can discover reasoning patterns through trial and error, though less efficiently than explicit teaching.

The interaction between training stages reveals complementary mechanisms: supervised learning with CoT establishes reasoning frameworks, while reinforcement learning optimizes their application. Models receiving both achieve consistently superior performance, validating our three-stage approach.

## 5 RELATED WORK

### 5.1 LLMs FOR RESEARCH ASSISTANCE

Domain-tuned language models such as SciBERT (Beltagy et al., 2019) established early gains for scholarly text. More recent systems extend beyond embeddings toward research assistance: the *AI Scientist* series automates ideation and paper drafting (Lu et al., 2024; Yamada et al., 2025), Google's *AI Co-Scientist* coordinates Gemini-based agents for experimental design (Gottweis et al., 2025), and OpenAI's *deep research* agent enables multi-step literature reviews (OpenAI, 2025a). These efforts highlight the utility of LLMs for research workflows, but they focus on productivity and coverage rather than the argumentative structure that underpins scientific reasoning.

### 5.2 ARGUMENTATION IN SCHOLARLY DISCOURSE

Work on discourse in science has emphasized rhetorical structure and argument mining. Argumentative zoning mapped rhetorical roles in articles (Teufel & Moens, 1999; Teufel et al., 1999), and tools such as ArguminSci extracted claims and evidence (Lauscher et al., 2018). More recently, dialogue-

oriented datasets have been introduced, including ArgSciChat for expert discussions (Ruggeri et al., 2023) and PaperPersiChat for chatbot-guided dialogue (Chernyavskiy et al., 2023). Outside scholarly domains, corpora such as IBM Debater (Slonim et al., 2021), Perspectrum (Chen et al., 2019), and DebateSum (Roush & Balaji, 2020) target news or social media. While these resources provide valuable foundations, they do not capture the methodological critique, evidential standards, or multi-level coherence that define scientific argumentation.

### 5.3 REASONING FOR SCHOLARLY TASKS

Recent work has shown that large models struggle with scientific verification and critique. On SCI-FACT, smaller domain-tuned models outperform frontier LLMs (Wadden et al., 2020); failures also appear in causal verification (CHECKWHY, Si et al. 2024) and peer review analysis (Du et al., 2024). Explicit reasoning supervision has been effective in other domains: chain-of-thought improves mathematical and multimodal science QA (Wei et al., 2022; Lu et al., 2022); synthetic rationales benefit smaller models (Magister et al., 2023; Wang et al., 2024); and structured reasoning approaches such as MULTIVERS (Wadden et al., 2022) and STRIVE (Gong et al., 2025) outperform flat pipelines. Interactive approaches like critique (Du et al., 2023) and debate (Hegazy & Wedel, 2024) further highlight the value of reasoning scaffolds. However, these techniques have been evaluated mostly on mathematics, logic, or short fact verification, leaving open whether they extend to the discourse-level demands of scientific argumentation.

## 6 CONCLUSION & FUTURE WORK

This work presents the first systematic evaluation of language model capabilities on scientific argumentation, revealing substantial gaps despite strong general reasoning abilities. Our key insight is that effective scientific argumentation requires training approaches that explicitly scaffold reasoning processes within domain-specific argumentative conventions, not general reasoning alone. The substantial performance gains across all tasks and competitive performance with o3-class systems provide strong evidence for this claim.

Our findings have immediate implications for AI-assisted research. The demonstrated abilities to identify claims, evaluate evidence, and generate coherent rebuttals, combined with out-of-domain generalization to conversational setting, suggest specialized reasoning training can produce models capable of meaningful participation in scientific discourse beyond task-specific pattern matching.

Several limitations constrain generalizability: evaluation occurs only within scientific domains, focuses on computational tasks that may not capture authentic discourse complexity, and relies on automatic metrics that may miss subtle argument quality aspects important to human evaluators. Future work should extend evaluation to additional argumentative domains, incorporate human evaluation studies, and investigate integration with broader research workflows.

## 7 REPRODUCIBILITY STATEMENT

We release all evaluation protocols, reasoning traces, and trained model checkpoints to support future research. We share used hyperparameters and seeds in training in C. As AI systems increasingly assist in scientific research, ensuring their capacity for rigorous argumentative reasoning becomes essential for maintaining scientific integrity and advancement.

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

## A ABLATION RESULTS

### A.1 REWARD FUNCTION ABLATIONS

| Model | Claim Detection | | Evidence Detection | | Evidence Classification | | Stance Detection | |
|---|---|---|---|---|---|---|---|---|
| | Acc | $F_1$ | Acc | $F_1$ | Acc | $F_1$ | Acc | $F_1$ |
| All | 84.75 | 82.48 | 71.00 | **43.23** | **68.75** | **71.06** | 53.25 | 54.88 |
| w/o ans | 61.50 | 66.67 | 51.50 | 22.66 | 7.50 | 12.20 | 34.00 | 16.92 |
| w/o attr | 59.00 | 83.41 | **99.50** | 33.25 | 43.00 | 44.19 | 55.50 | 50.72 |
| w/o fmt | 75.50 | 85.27 | 94.50 | 32.39 | 58.50 | 56.77 | **59.00** | **57.31** |
| w/o faith | **99.00** | **95.43** | **99.50** | 33.25 | 62.50 | 49.66 | 58.00 | 55.43 |

Table 3: Reward component ablation for argument mining tasks. "All" uses all reward components; subsequent rows remove individual components: ans (answer accuracy $r_{ans}$), attr (attribution $r_{attr}$), fmt (format compliance $r_{fmt}$), and faith (faithfulness $r_{faith}$). Green/red underlines indicate improvements/degradations relative to "All". Removing faithfulness constraints paradoxically improves some metrics, revealing models exploit evaluation artifacts rather than develop genuine reasoning. Highest scores bolded.

### A.2 CoT ABLATION RESULTS

## B DATA COLLECTION

**System Persona Variants.** We rotate between three teacher personas: *"You are a strict scientific teacher who writes short, grounded rationales"*, *"You are a meticulous scientific grader focusing on*

| Model | ARIES | | | PRRCA | | | QASA-FF | | |
|---|---|---|---|---|---|---|---|---|---|
| | ROUGE-L | METEOR | BERT | ROUGE-L | METEOR | BERT | ROUGE-L | METEOR | BERT |
| All | 14.75 | 13.65 | 83.47 | 14.21 | **18.01** | 84.06 | 16.95 | 15.32 | 82.47 |
| w/o ans | 7.65 | 4.53 | 79.65 | 10.94 | 9.32 | 78.61 | 16.10 | 21.66 | 75.63 |
| w/o attr | 19.54 | 19.58 | **85.11** | 16.29 | 17.15 | 84.38 | 13.22 | 23.41 | 79.86 |
| w/o fmt | **19.68** | 19.29 | 84.76 | **17.92** | 17.88 | 83.81 | 14.46 | 25.22 | 77.68 |
| w/o faith | **20.68** | **20.01** | 84.84 | 17.17 | 16.51 | **85.17** | **22.88** | **26.42** | **84.85** |

Table 4: Reward component ablation for generation tasks. Table formatting follows the same rules in Table 3.

*concise, text-grounded reasoning"*, and *"You are a careful science coach who values precise, well-grounded explanations"*. Each persona encourages different reasoning styles while maintaining scientific rigor.

**Instruction Template Variants.** Task instructions vary deterministically across alternative phrasings. For stance detection, we alternate between *"Task: Determine the stance of the rebuttal sentence toward the review excerpt"* and *"Question: What is the rebuttal's stance with respect to the review?"* Rule specifications also vary, including different formulations of format requirements and reasoning constraints.

### B.1    EXAMPLE PROMPTS

#### B.1.1    REBUTTAL GENERATION

```
System:
You are a meticulous scientific grader focusing on concise,
text-grounded reasoning.
```

```
Prompt
Task:  Write a rebuttal to the reviewer using ONLY the snippets.  In
reasoning, briefly plan your stance (agree/clarify/counter), cite
snippet NUMBERS (e.g., #1, #3), and outline 2{3 actions.  Do NOT copy
long text; cite numbers.

Reviewer comment:"The problem and motivation for the paper is weak.
The introduction does not provide any reason as to why this is an
important problem or approach that needs to be considered important
by the NLP research community."
Snippets:
```

| Model | Claim Detection | | Evidence Detection | | Evidence Classification | | Stance Detection | |
|---|---|---|---|---|---|---|---|---|
| | Acc | $F_1$ | Acc | $F_1$ | Acc | $F_1$ | Acc | $F_1$ |
| Base (1B) | 12.00 | 52.07 | 19.50 | 10.88 | 28.50 | 32.68 | 21.50 | 23.23 |
| SFT+CoT | 93.50 | 97.61 | 95.00 | 32.48 | **65.00** | **60.72** | 19.50 | 24.86 |
| SFT-CoT | 48.25 | 51.89 | 97.50 | 32.91 | 59.75 | 60.21 | 19.50 | 25.91 |
| RL+CoT | **95.50** | **98.22** | **99.50** | **33.25** | 47.00 | 43.16 | **51.50** | **46.71** |
| RL-CoT | 41.50 | 48.06 | 74.50 | 28.46 | 47.00 | 57.32 | 20.00 | 28.24 |

Table 5: Chain-of-thought ablation for argument mining tasks. Base represents the untrained model. We compare training with (+CoT) and without (-CoT) chain-of-thought reasoning across both SFT and RL stages. Green/red underlines indicate better/worse performance without CoT. Acc denotes accuracy. Values represent means across test sets with highest scores bolded.

```
#1.  1.  --- PAPER BODY (truncated) --- (INTRODUCTION) Recent years
have seen a revolution in Natural Language Processing (NLP) thanks
to the advances in machine learning.  2.  Finally, we offer detailed
analysis and discussion, which should be useful for researchers aiming
to explore the utility of contrastive loss in NLP.  --- PAPER BODY
(truncated) --- (INTRODUCTION) Recent years have seen a revolution in
Natural Language Processing (NLP) thanks to the advances in machine
learning.  3.  Example-based shuffling finds a data order similar to
the initial one, and the quality does not degrade much.  (CQA-B) The
results for CQA-B are shown in Table 2 .  Again, we did not use the
question position, which is a critically important feature for the
best systems.  4.  Our experimental results show sizable improvements
on a number of datasets and pairwise sentence scoring tasks including
classification, ranking, and regression.  Finally, we offer detailed
analysis and discussion, which should be useful for researchers aiming
to explore the utility of contrastive loss in NLP.  --- PAPER BODY
(truncated) --- (INTRODUCTION) Recent years have seen a revolution in
Natural Language Processing (NLP) thanks to the advances in machine
learning.
Respond with <reasoning> then <answer>.
<reasoning>
Gold answer:  Thank you for your feedback.  We understand that the
introduction needs to more clearly articulate the problem being
addressed and the motivation for our approach.  The paragraph you've
highlighted is a summary of one of our contributions and does not
contain the problem statement or motivation you requested.  We will
revise the introduction to include a dedicated section or paragraphs
that explain the importance of learning better task-specific sentence
embeddings for pairwise tasks and why exploring contrastive loss,
particularly in the ways we propose, is a significant direction for
NLP research.
```

**Teacher answer**

```
Thank you for your feedback.  We understand that the introduction
needs to more clearly articulate the problem being addressed and
the motivation for our approach.  The paragraph you've highlighted
is a summary of one of our contributions and does not contain the
problem statement or motivation you requested.  We will revise the
introduction to include a dedicated section or paragraphs that explain
the importance of learning better task-specific sentence embeddings for
pairwise tasks and why exploring contrastive loss, particularly in the
ways we propose, is a significant direction for NLP research.
```

| Model | ARIES | | | PRRCA | | | QASA-FF | | |
|---|---|---|---|---|---|---|---|---|---|
| | BERT | ROUGE-L | METEOR | BERT | ROUGE-L | METEOR | BERT | ROUGE-L | METEOR |
| Base (1B) | 41.00 | 4.00 | 2.00 | 84.00 | 12.00 | 8.00 | 86.17 | 18.79 | 19.16 |
| SFT+CoT | 85.49 | 20.54 | **22.34** | **85.53** | **18.40** | **19.54** | **86.85** | **19.62** | 21.50 |
| SFT w/o CoT | 85.47 | 20.36 | 21.67 | 85.49 | 18.17 | 19.28 | 43.43 | 10.08 | 10.69 |
| RL+CoT | 84.85 | 20.24 | 19.66 | 84.53 | 18.05 | 17.69 | 85.49 | 19.35 | 21.39 |
| RL w/o CoT | **85.57** | **22.24** | 20.61 | 84.83 | 17.03 | 15.32 | 78.28 | 13.58 | **24.44** |

Table 6: CoT ablation for generation tasks. Table formatting follows the same rules in Table 5.

**Student prompt**

```
Task:  Write a rebuttal to the reviewer using ONLY the snippets.  In
reasoning, briefly plan your stance (agree/clarify/counter), cite
snippet NUMBERS (e.g., #1, #3), and outline 2{3 actions.  Do NOT copy
long text; cite numbers.

Reviewer comment:"The problem and motivation for the paper is weak.
The introduction does not provide any reason as to why this is an
important problem or approach that needs to be considered important
by the NLP research community."
Snippets:
#1.  1.  --- PAPER BODY (truncated) --- (INTRODUCTION) Recent years
have seen a revolution in Natural Language Processing (NLP) thanks
to the advances in machine learning.  2.  Finally, we offer detailed
analysis and discussion, which should be useful for researchers aiming
to explore the utility of contrastive loss in NLP. --- PAPER BODY
(truncated) --- (INTRODUCTION) Recent years have seen a revolution in
Natural Language Processing (NLP) thanks to the advances in machine
learning.  3.  Example-based shuffling finds a data order similar to
the initial one, and the quality does not degrade much.  (CQA-B) The
results for CQA-B are shown in Table 2 .  Again, we did not use the
question position, which is a critically important feature for the
best systems.  4.  Our experimental results show sizable improvements
on a number of datasets and pairwise sentence scoring tasks including
classification, ranking, and regression.  Finally, we offer detailed
analysis and discussion, which should be useful for researchers aiming
to explore the utility of contrastive loss in NLP. --- PAPER BODY
(truncated) --- (INTRODUCTION) Recent years have seen a revolution in
Natural Language Processing (NLP) thanks to the advances in machine
learning.
Respond with <reasoning> then <answer>.
<reasoning>
```

### B.1.2 STANCE DETECTION

**System**

```
You are a careful science coach who values precise, well-grounded
explanations.
```

**Prompt**

```
Instruction:  Classify the rebuttal sentence's stance relative to the
review excerpt.
Choose exactly one of:  Concur, Dispute, Non-arg.

Review excerpt:
- in section 4.3, there is no guarantee that the intersection between
the training set and test set is empty.  - in section 4.3 how is the
reconstruction built (Figure 3b)?

Rebuttal sentence:
In other words, every valid output is still valid with rotation,
scaling and translation.

Format rules:
- Output ONLY two XML tags:  <reasoning> ...  </reasoning> then
<answer> ...  </answer>.
```

```
- <reasoning>:  2-4 sentences, paraphrased and grounded in the text.
No long quotes, no label words.
- <answer>:  EXACTLY the gold label/answer, nothing else.

Format:
<reasoning>
...
</reasoning>
<answer>
GOLD_LABEL
</answer>
Respond with <reasoning> then <answer>.
<reasoning>
Gold label:  CONCUR
```

**Teacher answer**
```
CONCUR
```

**Student prompt**
```
Instruction:  Classify the rebuttal sentence's stance relative to the
review excerpt.
Choose exactly one of:  Concur, Dispute, Non-arg.

Review excerpt:
- in section 4.3, there is no guarantee that the intersection between
the training set and test set is empty.  - in section 4.3 how is the
reconstruction built (Figure 3b)?

Rebuttal sentence:
In other words, every valid output is still valid with rotation,
scaling and translation.

Format rules:
- Output ONLY two XML tags:  <reasoning> ...  </reasoning> then
<answer> ...  </answer>.
- <reasoning>:  2-4 sentences, paraphrased and grounded in the text.
No long quotes, no label words.
- <answer>:  EXACTLY the gold label/answer, nothing else.

Format:
<reasoning>
...
</reasoning>
<answer>
GOLD_LABEL
</answer>
Respond with <reasoning> then <answer>.
<reasoning>
```

### B.1.3  CLAIM DETECTION

**System**
```
You are a meticulous scientific grader focusing on concise,
text-grounded reasoning.
```

**Prompt**
```
Task:  Identify which numbered sentences contain verifiable scientific
claims.

Abstract (numbered):
Research in two-dimensional atomic crystals has recently focused on
their heterostructures, and the advancements in this emerging field has
already led to fascinating discoveries such as superconductivity and
magnetism.  [1] However, thousands of different 2D layered materials
and their permutations amount to almost infinite heterostructure
combinations.  [2] This research will develop a novel ML-guided
DFT framework, in conjunction with physically motivated atomistic
descriptors, which applies data science in the search for designer
heterostructures with targeted properties.  [3] As a proof-of-concept,
we will demonstrate heterostructures with perfect light absorption
through optimizing the band nesting between the filled and empty
bands as well as giant piezoelectricity through engineering the
electronegativity dipole moments.  [4] These heterostructures
identified with the targeted properties will be grown with ultra-clean
state-of-the-art MBE approaches, and their absorption and piezoelectric
coefficients characterized.  [5] Corroboration between experiments and
theory will then instruct on possible improvements to the proposed ML
and DFT models and overall strategy.  [6] The successful demonstration
of these new designer 2D heterostructures would usher in a new era of
efficient and purposeful materials design methodology.

Output policy:
- Produce two blocks only (<reasoning>, <answer>).
- Keep <reasoning> concise (2-4 sentences), grounded, paraphrased, and
free of label terms.
- <answer> must match the gold label/answer exactly.
Notes for <reasoning>:
- Give a one-line definition of a verifiable claim.
- Explain criteria briefly (e.g., testable mechanism, measurable
effect).
Gold answer (comma-separated IDs or 'none'):  0,1

Gold answer:  0,1
Return exactly:
<reasoning>
...
</reasoning>
<answer>
GOLD_LABEL
</answer>
Respond with <reasoning> then <answer>.
<reasoning>
```

**Teacher answer**
```
0,1
```

**Student prompt**
```
Task:  Identify which numbered sentences contain verifiable scientific
claims.
```

```
Abstract (numbered):
Research in two-dimensional atomic crystals has recently focused on
their heterostructures, and the advancements in this emerging field has
already led to fascinating discoveries such as superconductivity and
magnetism.  [1] However, thousands of different 2D layered materials
and their permutations amount to almost infinite heterostructure
combinations.  [2] This research will develop a novel ML-guided
DFT framework, in conjunction with physically motivated atomistic
descriptors, which applies data science in the search for designer
heterostructures with targeted properties.  [3] As a proof-of-concept,
we will demonstrate heterostructures with perfect light absorption
through optimizing the band nesting between the filled and empty
bands as well as giant piezoelectricity through engineering the
electronegativity dipole moments.  [4] These heterostructures
identified with the targeted properties will be grown with ultra-clean
state-of-the-art MBE approaches, and their absorption and piezoelectric
coefficients characterized.  [5] Corroboration between experiments and
theory will then instruct on possible improvements to the proposed ML
and DFT models and overall strategy.  [6] The successful demonstration
of these new designer 2D heterostructures would usher in a new era of
efficient and purposeful materials design methodology.

Output policy:
- Produce two blocks only (<reasoning>, <answer>).
- Keep <reasoning> concise (2-4 sentences), grounded, paraphrased, and
free of label terms.
- <answer> must match the gold label/answer exactly.
Notes for <reasoning>:
- Give a one-line definition of a verifiable claim.
- Explain criteria briefly (e.g., testable mechanism, measurable
effect).
Gold answer (comma-separated IDs or 'none'):  0,1

Return exactly:
<reasoning>
...
</reasoning>
<answer>
GOLD_LABEL
</answer>
Respond with <reasoning> then <answer>.
<reasoning>
```

### B.1.4 META-REVIEW GENERATION

**System**
```
You are a meticulous scientific grader focusing on concise,
text-grounded reasoning.
```

**Prompt**
```
Task:  Write a concise meta-review using ONLY the snippets.
In reasoning, list 2-4 key claims as C1:/C2:  with who
supports/contradicts, citing snippet NUMBERS (e.g., #2, #4-#5); end
with a one-line recommendation rationale.  Do NOT copy long text; cite
numbers.
```

```
Paper title:  Learning what you can do before doing anything

Snippets:
#1.  1.  Learning what you can do before doing anything The authors
propose a way to learn models that predict what will happen next in
scenarios where action-labels are not available in abundance.  2.  It
only becomes clear in the technical and experimental section what
problem is addressed in this paper.  3.  It only becomes clear in the
technical and experimental section what problem is addressed in this
paper.  Rebuttal:  4.  The abstract and introduction are too vague
and general.  It only becomes clear in the technical and experimental
section what problem is addressed in this paper.
Respond with <reasoning> then <answer>.
<reasoning>
Gold answer:  The reviewers had some concerns regarding clarity and
evaluation but in general liked various aspects of the paper.  The
authors did a good job of addressing the reviewers' concerns so
acceptance is recommended.
```

**Teacher answer**

```
The reviewers had some concerns regarding clarity and evaluation but in
general liked various aspects of the paper.  The authors did a good job
of addressing the reviewers' concerns so acceptance is recommended.
```

**Student prompt**

```
Task:  Write a concise meta-review using ONLY the snippets.
In reasoning, list 2-4 key claims as C1:/C2:  with who
supports/contradicts, citing snippet NUMBERS (e.g., #2, #4-#5); end
with a one-line recommendation rationale.  Do NOT copy long text; cite
numbers.

Paper title:  Learning what you can do before doing anything

Snippets:
#1.  1.  Learning what you can do before doing anything The authors
propose a way to learn models that predict what will happen next in
scenarios where action-labels are not available in abundance.  2.  It
only becomes clear in the technical and experimental section what
problem is addressed in this paper.  3.  It only becomes clear in the
technical and experimental section what problem is addressed in this
paper.  Rebuttal:  4.  The abstract and introduction are too vague
and general.  It only becomes clear in the technical and experimental
section what problem is addressed in this paper.
Respond with <reasoning> then <answer>.
<reasoning>
```

### B.1.5 SYSTEMATIC ANSWER COMPOSITION

**System**

```
You are a strict scientific teacher who writes short, grounded
rationales.
```

**Prompt**

```
1080
1081   You are a scientific QA teacher.  Read the passage and answer
1082   concisely.
       In <reasoning> (3{5 sentences), justify the answer with key details
1083   from the passage.  Ground your explanation; avoid long verbatim
1084   copying.
1085   Gold answer:  Metrics used for comparison are AP , multi-scale
1086   train/test, horizontal flip test, and online hard example mining
1087   (OHEM).
1088   In <answer>, output EXACTLY the provided gold answer string.
1089
1090   Passage:
1091   We perform a thorough comparison of Mask R-CNN to the state of the art
       along with comprehensive ablations on the COCO dataset [28].  We report
1092   the standard COCO metrics including AP (averaged over IoU thresholds),
1093   AP{}_{50}, AP{}_{75}, and AP{}_{S}, AP{}_{M}, AP{}_{L} (AP at different
1094   scales).  Unless noted, AP is evaluating using mask IoU. As in previous
1095   work [5, 27], we train using the union of 80k train images and a
1096   35k subset of val images (trainval35k), and report ablations on the
1097   remaining 5k val images (minival).  We also report results on test-dev
1098   [28].
1099   ---
1100   We compare Mask R-CNN to the state-of-the-art methods in instance
1101   segmentation in Table 1.  All instantiations of our model outperform
1102   baseline variants of previous state-of-the-art models.  This includes
1103   MNC [10] and FCIS [26], the winners of the COCO 2015 and 2016
       segmentation challenges, respectively.  Without bells and whistles,
1104   Mask R-CNN with ResNet-101-FPN backbone outperforms FCIS+++ [26], which
1105   includes multi-scale train/test, horizontal flip test, and online hard
1106   example mining (OHEM) [38].  While outside the scope of this work, we
1107   expect many such improvements to be applicable to ours.
1108
1109   Question:
1110   What metrics should be used for comparison of Mask R-CNN to the state
1111   of the art on the COCO dataset ?
1112
1113   Rules:
1114   - Return EXACTLY two XML blocks:  <reasoning> then <answer>.
       - In <reasoning>:  3{5 sentences, grounded in the provided text; it's
1115   fine to use key terms or short phrases, but avoid long verbatim copying
1116   or listing options.
1117   - In <answer>:  output EXACTLY the provided gold label/answer.  No
1118   extra text.
1119
1120   Output:
1121   <reasoning>
1122   ...
1123   </reasoning>
1124   <answer>
1125   GOLD_LABEL
1126   </answer>
       Respond with <reasoning> then <answer>.
1127   <reasoning>
1128
1129
1130   Teacher answer
1131   Metrics used for comparison are AP , multi-scale train/test, horizontal
1132   flip test, and online hard example mining (OHEM).
1133
```

**Student prompt**

```
You are a scientific QA teacher.  Read the passage and answer
concisely.
In <reasoning> (3{5 sentences), justify the answer with key details
from the passage.  Ground your explanation; avoid long verbatim
copying.
In <answer>, output EXACTLY the provided gold answer string.

Passage:
We perform a thorough comparison of Mask R-CNN to the state of the art
along with comprehensive ablations on the COCO dataset [28].  We report
the standard COCO metrics including AP (averaged over IoU thresholds),
AP{}_{50}, AP{}_{75}, and AP{}_{S}, AP{}_{M}, AP{}_{L} (AP at different
scales).  Unless noted, AP is evaluating using mask IoU. As in previous
work [5, 27], we train using the union of 80k train images and a
35k subset of val images (trainval35k), and report ablations on the
remaining 5k val images (minival).  We also report results on test-dev
[28].
---
We compare Mask R-CNN to the state-of-the-art methods in instance
segmentation in Table 1.  All instantiations of our model outperform
baseline variants of previous state-of-the-art models.  This includes
MNC [10] and FCIS [26], the winners of the COCO 2015 and 2016
segmentation challenges, respectively.  Without bells and whistles,
Mask R-CNN with ResNet-101-FPN backbone outperforms FCIS+++ [26], which
includes multi-scale train/test, horizontal flip test, and online hard
example mining (OHEM) [38].  While outside the scope of this work, we
expect many such improvements to be applicable to ours.

Question:
What metrics should be used for comparison of Mask R-CNN to the state
of the art on the COCO dataset ?

Rules:
- Return EXACTLY two XML blocks:  <reasoning> then <answer>.
- In <reasoning>:  3{5 sentences, grounded in the provided text; it's
fine to use key terms or short phrases, but avoid long verbatim copying
or listing options.
- In <answer>:  output EXACTLY the provided gold label/answer.  No
extra text.

Output:
<reasoning>
...
</reasoning>
<answer>
GOLD_LABEL
</answer>
Respond with <reasoning> then <answer>.
<reasoning>
```

### B.1.6 EVIDENCE CLASSIFICATION

**System**
You are a meticulous scientific grader focusing on concise, text-grounded reasoning.

---

**Prompt**
Question: Does the evidence SUPPORT or REFUTE the claim?

Claim:
General exercise therapy is more effective than scapular stabilizer exercises in reducing pain and improving function of the shoulder.

Evidence:
CONCLUSION A specific exercise strategy, focusing on strengthening eccentric exercises for the rotator cuff and concentric/eccentric exercises for the scapula stabilisers, is effective in reducing pain and improving shoulder function in patients with persistent subacromial impingement syndrome.

Choose one of: REFUTES, SUPPORTS.

Rules:
- Return EXACTLY the two XML blocks.
- In <reasoning>: 2-4 sentences, paraphrased, grounded ONLY in provided text; DO NOT copy long spans; DO NOT include label words (SUPPORTS, REFUTES, SUPPORT, CONTRADICT, NEI, Concur, Dispute, Non-arg, NON-ARG, REFUTE, CONTRADICTS).
Gold label: REFUTES
- In <answer>: output EXACTLY the provided gold label/answer. No extra text.

Your output must be exactly:
<reasoning>
...
</reasoning>
<answer>
GOLD_LABEL
</answer>
Respond with <reasoning> then <answer>.
<reasoning>

---

**Teacher answer**
REFUTES

---

**Student prompt**
Question: Does the evidence SUPPORT or REFUTE the claim?

Claim:
General exercise therapy is more effective than scapular stabilizer exercises in reducing pain and improving function of the shoulder.

Evidence:
CONCLUSION A specific exercise strategy, focusing on strengthening

```
eccentric exercises for the rotator cuff and concentric/eccentric
exercises for the scapula stabilisers, is effective in reducing pain
and improving shoulder function in patients with persistent subacromial
impingement syndrome.

Choose one of:  REFUTES, SUPPORTS.

Rules:
- Return EXACTLY the two XML blocks.
- In <reasoning>:  2-4 sentences, paraphrased, grounded ONLY in
provided text; DO NOT copy long spans; DO NOT include label words
(SUPPORTS, REFUTES, SUPPORT, CONTRADICT, NEI, Concur, Dispute, Non-arg,
NON-ARG, REFUTE, CONTRADICTS).
- In <answer>:  output EXACTLY the provided gold label/answer.  No
extra text.

Your output must be exactly:
<reasoning>
...
</reasoning>
<answer>
GOLD_LABEL
</answer>
Respond with <reasoning> then <answer>.
<reasoning>
```

### B.1.7 EVIDENCE DETECTION

**System**
```
You are a careful science coach who values precise, well-grounded
explanations.
```

---

**Prompt**
```
Task:  Decide if the sentence can support or contradict the claim, or
if it's NEI. Choose one of:  NEI, CONTRADICT, SUPPORT.

Claim:
Cytochrome c is released from the mitochondrial intermembrane space to
cytosol during apoptosis.

Sentence:
BACKGROUND & AIMS Helicobacter pylori-induced gastric epithelial cell
(GEC) apoptosis is a complex process that includes activation of the
tumor suppressor p53.

Rules:
- Return EXACTLY the two XML blocks.
- In <reasoning>:  2-4 sentences, paraphrased, grounded ONLY in
provided text; DO NOT copy long spans; DO NOT include label words
(SUPPORTS, REFUTES, SUPPORT, CONTRADICT, NEI, Concur, Dispute, Non-arg,
NON-ARG, REFUTE, CONTRADICTS).
Gold label:  NEI
- In <answer>:  output EXACTLY the provided gold label/answer.  No
extra text.

Output:
```

```
<reasoning>
...
</reasoning>
<answer>
GOLD_LABEL
</answer>
Respond with <reasoning> then <answer>.
<reasoning>
```

**Teacher answer**
```
NEI
```

**Student prompt**
```
Task:  Decide if the sentence can support or contradict the claim, or
if it's NEI. Choose one of:  NEI, CONTRADICT, SUPPORT.

Claim:
Cytochrome c is released from the mitochondrial intermembrane space to
cytosol during apoptosis.

Sentence:
BACKGROUND & AIMS Helicobacter pylori-induced gastric epithelial cell
(GEC) apoptosis is a complex process that includes activation of the
tumor suppressor p53.

Rules:
- Return EXACTLY the two XML blocks.
- In <reasoning>:  2-4 sentences, paraphrased, grounded ONLY in
provided text; DO NOT copy long spans; DO NOT include label words
(SUPPORTS, REFUTES, SUPPORT, CONTRADICT, NEI, Concur, Dispute, Non-arg,
NON-ARG, REFUTE, CONTRADICTS).
- In <answer>:  output EXACTLY the provided gold label/answer.  No
extra text.

Output:
<reasoning>
...
</reasoning>
<answer>
GOLD_LABEL
</answer>
Respond with <reasoning> then <answer>.
<reasoning>
```

Table 7: Label distribution for `disapere` (Train/Dev/Test)

| Label | Train Count | Train Percent | Test Count | Test Percent |
|-------|-------------|---------------|------------|--------------|
| Dispute | 175 | 35.00% | 69 | 34.50% |
| Concur | 165 | 33.00% | 63 | 31.50% |
| Non-arg | 160 | 32.00% | 68 | 34.00% |
| Total | 500 | 100.00% | 200 | 100.00% |

Table 8: Label distribution for `scifact_cls` (Train/Dev/Test)

| Label | Train Count | Train Percent | Test Count | Test Percent |
|---|---|---|---|---|
| SUPPORTS | 150 | 60.00% | 126 | 63.00% |
| REFUTES | 100 | 40.00% | 74 | 37.00% |
| Total | 250 | 100.00% | 200 | 100.00% |

Table 9: Label distribution for `scifact_evdet` (Train/Dev/Test)

| Label | Train Count | Train Percent | Test Count | Test Percent |
|---|---|---|---|---|
| NEI | 246 | 98.40% | 196 | 98.00% |
| SUPPORT | 3 | 1.20% | 2 | 1.00% |
| CONTRADICT | 1 | 0.40% | 2 | 1.00% |
| Total | 250 | 100.00% | 200 | 100.00% |

## C  TRAINING SETUP

### C.1  SUPERVISED FINE-TUNING CONFIGURATION

We fine-tune Llama 3.2 models using the following configuration across training seeds 42, 156, 248:

| Hyperparameter | SFT | RL |
|---|---|---|
| Learning rate | 3e-5 (linear warmup) | 1e-6 (cosine annealing) |
| Training steps | 3,000 | 1,000 |
| Context length | 3,072 | 1,024 |
| Batch size | 32 (8 per device $\times$ 4 accumulation) | |
| Optimizer | AdamW ($\beta_1 = 0.9$, $\beta_2 = 0.999$, wd=0.01) | |
| Mixed precision | FP16 | |
| Group size | – | 8 generations per prompt |
| KL penalty | – | Adaptive $\beta$ scheduling |
| Reward normalization | – | Per-batch standardization |
| GRPO rollouts | – | 512 |

Table 10: Training hyperparameters for supervised fine-tuning (SFT) and reinforcement learning (RL) stages.

**Adaptive KL Constraint.** We implement adaptive $\beta$ scheduling:

$$\beta_{t+1} = \begin{cases} \min(\beta_t \cdot 1.5, \beta_{\max}) & \text{if } \text{KL}_t > (1 + \epsilon) \cdot \text{KL}_{\text{target}} \\ \max(\beta_t / 1.5, \beta_{\min}) & \text{if } \text{KL}_t < (1 - \epsilon) \cdot \text{KL}_{\text{target}} \\ \beta_t & \text{otherwise} \end{cases} \tag{4}$$

with $\text{KL}_{\text{target}} = 0.03$, $\epsilon = 0.2$, $\beta_{\min} = 0.02$, and $\beta_{\max} = 0.15$.

