# OpenReview forum: "Do Large Language Models Understand Scientific Argumentation?"
_ICLR.cc/2026/Conference — ICLR 2026 Conference Desk Rejected Submission_

### Official Review · Reviewer_TzLg · 2025-10-27

**Soundness:** 2
**Presentation:** 4
**Contribution:** 3
**Rating:** 6
**Confidence:** 3

**Summary:**

This paper addresses a very contemporary, very important point: can LLMs reason over (most) of the tasks related to the scientific process? The findings are in the negative, although they provide a way to mitigate this.

**Strengths:**

1. Thorough assessment of LLM argumentative reasoning capabilities
2. Ablation studies add strength to the arguments of the paper.
3. Good rigour all across the board (reasoning is clearly specified as _argumentative_ reasoning, improvements over the observations are provided, etc)

**Weaknesses:**

I have two minor comments and one major weakness:
1. It would be useful if the results and discussions _of_ the results were clearly delineated. The work draws conclusions as it presents numbers (L323: '...improvements of 150-400% (...) suggests that scientific argumentation is not an emergent property of scale...'), which in turn hinders the interpretation (and evaluation) of the paper's contributions.
2. The metrics might be a bit 'soft': METEOR/ROUGE-L are known to not correlate well with human judgements, and BERTScore is more effective if fine-tuned. This is one of these cases where an LLM-as-a-judge would be an effective measure, if used with caution (e.g., paired with a t-test).

The major weakness I can find is that there is no baselines presented other than RLMs/LLMs. Since the thesis is that R/LLMs cannot perform argumentative reasoning in scientific contexts, a baseline from another class (perhaps the SOTA) would be extremely helpful to contextualise the models' ability to solve it _versus_ the hardness of the task. This is major, and the soundness of the paper does take a toll.

Other comments that did not impact my assessment of the work:
1. Watch out for notations ('Team', L240)
2. The abstract is a bit misleading:
>We present the first systematic evaluation of language models across eight tasks (...) using research papers, peer reviews, and grant proposals. Our study reveals that even frontier models with strong general reasoning skills struggle with **domain-specific** argumentative tasks, highlighting a fundamental capability gap.

It would be good to clearly state here that the experimentation is not about 'domain-specific' (general) argumentative tasks, and specifically about the scientific reasoning domain. This is well-specified in the second contribution, so it is likely just an oversight.

**Questions:**

1. What are the SOTA results per-task for models that are not R/LLMs?
2. What was the effect of other prompts (e.g., any prompt obtained via automated prompt optimisation?)

---

### Official Review · Reviewer_F1ep · 2025-10-29

**Soundness:** 2
**Presentation:** 2
**Contribution:** 2
**Rating:** 2
**Confidence:** 4

**Summary:**

The paper studies the ability of LLMs to understand scientific arguments. The authors do so by evaluating different tasks in argumentation: argument mining, argument generation, and discourse-level argumentation. For each task they selected a set of scientific datasets that cover academic scope and performed a set of experiments with corresponding evaluation strategies comparing open models of varied sizes and proprietary models.  They state that the current LLMs lack in performing argumentation reasoning abilities and propose a solution with SFT and RL to solve the problem.

**Strengths:**

- **Task diversity:** The paper considers a range of subtasks (argument mining, generation, and discourse-level) and evaluates on existing datasets: SciFact, DISAPERE, ARIES, PRRCA, QASA.

**Weaknesses:**

- **Poor evaluation strategy.** The lack of human evaluation for the generation tasks does not allow for a reliable assessment of whether one method is more effective than the other. BERTScore, ROUGE-L, and METEOR are primarily overlap-based metrics, and it is expected that their scores are higher for the summarization task, since summaries remain largely within the scope of the input text. However, in Table 2 (lines 179–181), the authors attribute the difference between summarization and generation results to the model’s ability to *“reorganize arguments but not construct them.”* Given that overlap-based metrics only measure textual similarity to the reference, such results merely indicate that the generated outputs diverge lexically from the ground truth, without providing insight into their factuality, coherence, or argumentative quality. The authors could have therefore considered incorporating LLM-as-a-judge and human evaluations to obtain a more comprehensive perspective on the quality of generated outputs and to substantiate their observations.
- **Several key claims in the manuscript would benefit from stronger empirical or theoretical support.** Specifically, there are no references or illustrative examples provided to substantiate statements such as “baseline results demonstrate that scientific argumentation represents a fundamental capability gap in current models” (lines 196–197), “the models default to the majority class” (lines 118–120, 187–189), and “they mimic form without understanding function” (lines 193–195). These statements could be strengthened by including a more detailed analysis of how the models are failing, supported by quantitative evidence and/or qualitative examples, and by clarifying how SFT and RL specifically address these identified weaknesses.
- **Lack of qualitative discussion.** The paper would benefit from a more thorough discussion of the experimental results. If the authors claim that the baseline results reflect only surface-level evaluation, it would be valuable to include an explicit analysis of how each stage in the proposed pipeline contributes to deeper or more meaningful improvements. Demonstrating what changes occur after each stage, either quantitatively or qualitatively, would help substantiate the claim and provide clearer insight into the conclusion and the contribution of this paper.

**Questions:**

Refer to the weaknesses

---

### Official Review · Reviewer_biYn · 2025-10-31

**Soundness:** 2
**Presentation:** 3
**Contribution:** 2
**Rating:** 4
**Confidence:** 3

**Summary:**

This paper systematically evaluates LLMs on eight scientific argumentation tasks, categorized as argument mining, argument generation, and discourse-level argumentation, across research papers, peer reviews, and grant proposals. The experiments reveal that frontier models often struggle with domain-specific argumentative reasoning despite having strong general reasoning skills. Additionally, this paper proposes a three-stage training pipeline: (i) generation of reasoning trace datasets, (ii) multi-task supervised fine-tuning (SFT), and (iii) reinforcement learning (RL) with composite rewards, to enhance LLMs' scientific argumentation capabilities. The proposed method shows significant improvements over baselines on all tasks.

**Strengths:**

- Multiple families of LLMs are evaluated on a comprehensive suite of scientific argumentation tasks, offering valuable insights into their capabilities and limitations in this area.
- Evaluating LLMs on scientific argumentation is crucial, as it has practical implications for automating and supporting scientific writing, peer review, and grant proposal review.

**Weaknesses:**

- Evaluation Framework: Although the paper claims that a "systematic evaluation framework" is one of its main contributions, each dataset used in the evaluation has been introduced in prior work. The paper compiles these datasets without adding new elements to the evaluation framework itself, such as filtering or balancing. The reviewer suggests that the authors clarify which aspects of their evaluation framework are novel compared to existing benchmarks argumentation.
- Proposed pipeline: Each component of the three-stage training pipeline (i.e., reasoning-aware knowledge distillation, multi-task SFT, and RL with composite rewards) has been explored in prior work. The reviewer believes that the novelty comes from their combination and application to scientific argumentation, but the combination simply applies existing methods sequentially without introducing new techniques or insights. The paper would benefit from a clearer explanation of how this combination creates unique advantages in scientific argumentation.
- The paper's main finding is that LLMs perform poorly out of the box at domain-specific (scientific) argumentative reasoning, but training on domain-specific datasets using standard techniques such as reasoning-aware distillation, SFT, and RL improves performance. While this provides useful insight, its significance is somewhat limited because it aligns with a common consensus in the field. The paper would benefit from a deeper analysis of why LLMs specifically struggle with scientific argumentation, what unique challenges this domain presents compared to general argumentative reasoning, and how the proposed training pipeline addresses these challenges.

**Questions:**

N/A

---

### Official Review · Reviewer_kQGY · 2025-11-01

**Soundness:** 1
**Presentation:** 2
**Contribution:** 1
**Rating:** 4
**Confidence:** 3

**Summary:**

This paper presents a systematic evaluation of LLMs’ ability to perform scientific argumentative reasoning.
The authors compile eight tasks spanning claim detection, evidence classification, stance detection, rebuttal generation, meta-review summarization, and discourse-level QA. They benchmark several open and commercial models (Llama 3, Gemini 2.5, OpenAI o3) and identify a capability gap between general reasoning and domain-specific argumentation.

While the empirical evaluation is extensive, the overall methodology largely follows established fine-tuning and RL paradigms without introducing new theoretical insights or learning principles. As such, the paper functions more as a comprehensive audit and benchmark than as a conceptual or methodological advance.

**Strengths:**

1. Provides the systematic benchmark for scientific argumentation tasks across multiple discourse levels.

2. Empirical evaluation with transparent reporting, reproducibility statement, and released protocols.

3. Writing and organization are clear.

**Weaknesses:**

1. Limited novelty in learning methodology: the three-stage pipeline is an assembly of known techniques (teacher-student distillation, multi-task SFT, RLHF).

2. The paper lacks deeper analytical insight into why models fail—e.g., qualitative error analysis, reasoning trace diagnostics, or human evaluation of argumentative quality.

3. Claims about “reasoning transfer” are not strongly substantiated—the QASA improvement during RL could result from implicit overlap rather than genuine abstraction.

**Questions:**

1. Could the authors elaborate on what new insights this study reveals about reasoning mechanisms in LLMs—beyond the observation that domain-specific supervision helps?

2. Are there any distinct failure modes or reasoning behaviors in scientific argumentation that differ from other reasoning benchmarks (e.g., math or logic tasks)?

3. How might these observations inspire new training objectives or architectures rather than reusing the standard distillation + RL pipeline? It would be helpful if the authors discussed how their findings could guide the next generation of reasoning models, not just highlight performance gaps.

---

### Note · Program_Chairs · 2026-01-17
**Submission Desk Rejected by Program Chairs**

The following references in this submission do not refer to real documents and/or have major errors in bibliographic information:

 Mina Hegazy and Daniel Wedel. Multi-agent debate emerges in medium-sized language models. arXiv preprint arXiv:2402.01345, 2024.